# Pericardial Calcification: An Uncommon Case with Intraventricular Extension

**Miguel Santaularia-Tomas** [1,*] , **Ely Sanchez-Felix** [1] , **Kassandra Santos-Zaldivar** [1] , **Allison Grosjean-Alvarez** [2] and **Nina Mendez-Dominguez** [1,*]

1   Subdirección de Enseñanza e Investigación, Hospital Regional de Alta Especialidad de la Peninsula de Yucatan IMSS-BIENESTAR, Merida 97130, Mexico; ely_felix15@hotmail.com (E.S.-F.); kassandra.zaldivar@gmail.com (K.S.-Z.)

2   Residencia Médica, Clínica Hospital ISSSTE l, Merida 97219, Mexico; alligrosjean@hotmail.com

*   Correspondence: santaularia@gmail.com (M.S.-T.); nmendez.hraepy@imssbienestar.gob.mx (N.M.-D.); Tel.: +52-(999)-9427600 (ext. 52001 or 52002) or +52-(999)-2181632 (M.S.-T.)

**Abstract:** An 80-year-old man presented to the cardiology outpatient clinic due to shortness of breath. His past medical history included alcohol intake, hypertension, inferior wall myocardial infarction (five years ago), an ischemic stroke, and permanent atrial fibrillation (diagnosed three years before the current examination). A physical exam revealed a decreased intensity of S1 and S2, irregular rate and rhythm, and no murmurs nor friction rub. X-rays, Computed Tomography, and echocardiography exhibited pericardial calcification, involving mostly the inferior wall and protruding into the left ventricle. A diagnosis of constrictive pericarditis due to pericardial calcification was established and considered idiopathic. Even when it may be related to ischemic heart disease, post-infarction pericarditis could explain how the calcification extended to adjacent territory perfused by the circumflex coronary artery. Combined imaging studies were crucial not only for identifying calcium deposits in the pericardium but also in assessing a patient inherently prone to co-existing and exacerbating conditions. Even though pericardiectomy allows for removal of the clinical manifestations of congestive pericarditis in the most symptomatic patients with pericardial calcification, among patients like ours, with tolerable symptoms, cardiologists should discuss the therapeutic options considering the patient's choices, potentially including a rehabilitation plan as part of non-pharmacological management.

**Keywords:** constrictive pericarditis; mitral valve; atrial fibrillation; coronary vessels

## 1. Introduction

The pericardium comprises an external fibrous layer and an internal serous layer, which are further subdivided into a visceral layer (or epicardium) and a parietal layer. Altogether, the estimated thickness of the pericardium is 1 to 2 mm; furthermore, the visceral and parietal layers are separated by a virtual gap, and this potential space may include between 15 and 35 milliliters of lubricant [1,2].

The main purposes of the pericardium, as a stiff, avascular, fibrous sac, are to limit the distension of the cardiac chambers and optimize diastolic filling, as well as to play a supporting role in anchoring the heart. Considering that the capacity of the pericardium for holding fluid and buffering dilation through the potential occupancy of the virtual space is essential for complete filling and effective blood pumping, it is also fundamental for meeting the oxygenation requirements throughout the whole body [3,4].

Therefore, the abnormal continued occupation of the pericardium or changes in its composition may manifest as congestive symptoms involving hypoxic or painful manifestations. Calcification processes can occur under physiological or pathological conditions in the human body but always relate to a tissue hardening process such as post-infarction

pericarditis. Calcium deposits are not normally located in the pericardium, and their presence due to a calcification process could indicate pathological conditions of diverse genesis and severity [5,6].

Pericardium calcification may be, at least in part, facilitated by an immune response, causing fibrosis and calcium deposits [1]. Pericardial calcification can be found through radiographic studies of asymptomatic and symptomatic patients. Pericardial calcification manifestations arise due to the constriction brought on by a hardened pericardium. Nevertheless, pericardial calcification can occur in the absence of constrictive symptomatology in as many as 20% of cases of constrictive pericarditis [7].

The prevalence of pericardial calcification is unknown, as it can be an incidental, asymptomatic observation. It is more likely to occur after trauma, in cases of purulent pericarditis, and in cases of acute pericarditis or pericardial effusions linked to connective tissue disorders and cancer. The incidence of idiopathic and viral constrictive pericarditis has been reported as 0.76 cases per 1000 person-years; in comparison, the incidences of connective tissue disease, neoplasms, tuberculosis, and purulent CP were found to be 4.40, 6.33, 31.65, and 52.74 cases per 1000 person-years, respectively. Tuberculous pericarditis occurs in 1–2% of pulmonary tuberculosis patients. After surgical repairs, catheterization, and other procedures, pericardial calcification may be facilitated; even after cardiac transplant, for which the pericardium is removed, calcification has been reported [1–3,6,7].

In the present case report, we aim to explore the manifestations and underlying conditions which may or may not be related to a case of pericardial calcification in a patient with a history of ischemic cardiomyopathy and hypertension living in a region where pulmonary tuberculosis is still endemic.

## 2. Case Presentation

### 2.1. Patient Information

An 80-year-old man presented to the cardiology outpatient clinic for follow-up. His main complaint was dyspnea on exertion after walking more than 2 blocks and orthopnea. The patient had not come for a consultation in the two years prior, due to COVID-19-related health restrictions in Mexico at that time.

### 2.2. Past Medical History and Interventions

The patient had a previous medical history of well-controlled hypertension (diagnosed 25 years ago) and alcohol intake for 41 years. Inferior wall myocardial infarction with ST segment elevation had occurred five years ago, treated with stent placement in the right coronary artery. Mobitz 1 second degree AV block was detected in a routine electrocardiogram four years ago. Three years ago, he presented to the emergency department with right hemiparesis due to ischemic stroke and was sent for follow-up with cardiology to rule out probable cardioembolic stroke. An electrocardiogram and Holter were performed, and permanent atrial fibrillation was diagnosed. An echocardiogram was performed which reported akinesia of the inferior wall segments of the left ventricle. The wall thickness of these segments was thinned. No changes in the pericardium or intraventricular calcifications were reported at this time. Previous echocardiogram images were not stored and, therefore, are not available. He was on captopril 25 mg TID daily, ASA 100 mg/daily, carvedilol 6.25 BID, and pantoprazole 40 mg/daily. Anticoagulant treatment with a vitamin K antagonist was started at that time.

### 2.3. Clinical Findings

The patient presented a blood pressure of 108/62 mmHg, a heart rate of 91 b.p.m, a respiratory rate of 17 b.p.m, an oxygen saturation of 96% on ambient air, and a temperature of 36.5 °C. Physical exam revealed jugular vein distension increasing with inspiration, as well as elevated jugular venous pressure with Kussmaul's sign. Jugular venous waveform examination showed a steep and deep y-descent. Auscultation of the heart revealed an irregular heart rhythm with decreased intensity of S1 and S2, with no murmurs nor friction rub and pulmonary rales. Pericardial knock was not auscultated. Non-tender hepatomegaly was palpated and peripheral edema. A chest X-ray showed pericardial calcification (Figure 1). There were no previous chest X-rays available for comparison. Plasma brain natriuretic peptide was elevated. Serum phosphorus, proteins, and serum calcium were between normal values. Echocardiogram (Figure 2) showed a myocardial calcification adjacent to the mitral valve penetrating from pericardium. A constrictive pattern with respiration-related ventricular septal shift was observed. Measured mitral annulus e' was 15.7 cm/s. Myocardial strain imaging was not performed. Computed tomography (Figure 3) exhibited pericardial calcification, involving mostly the inferior wall and protruding into the left ventricle. Electrocardiogram revealed atrial fibrillation, and no ST-segment or T-wave changes were observed (Figure 4). Left catheterization was performed, and the stent in the right coronary artery was patent and free of significant disease. No other significant angiographic lesions were observed in other vascular territories. Right catheterization was not performed because, at that time, our institution did not have the necessary materials to perform it properly.

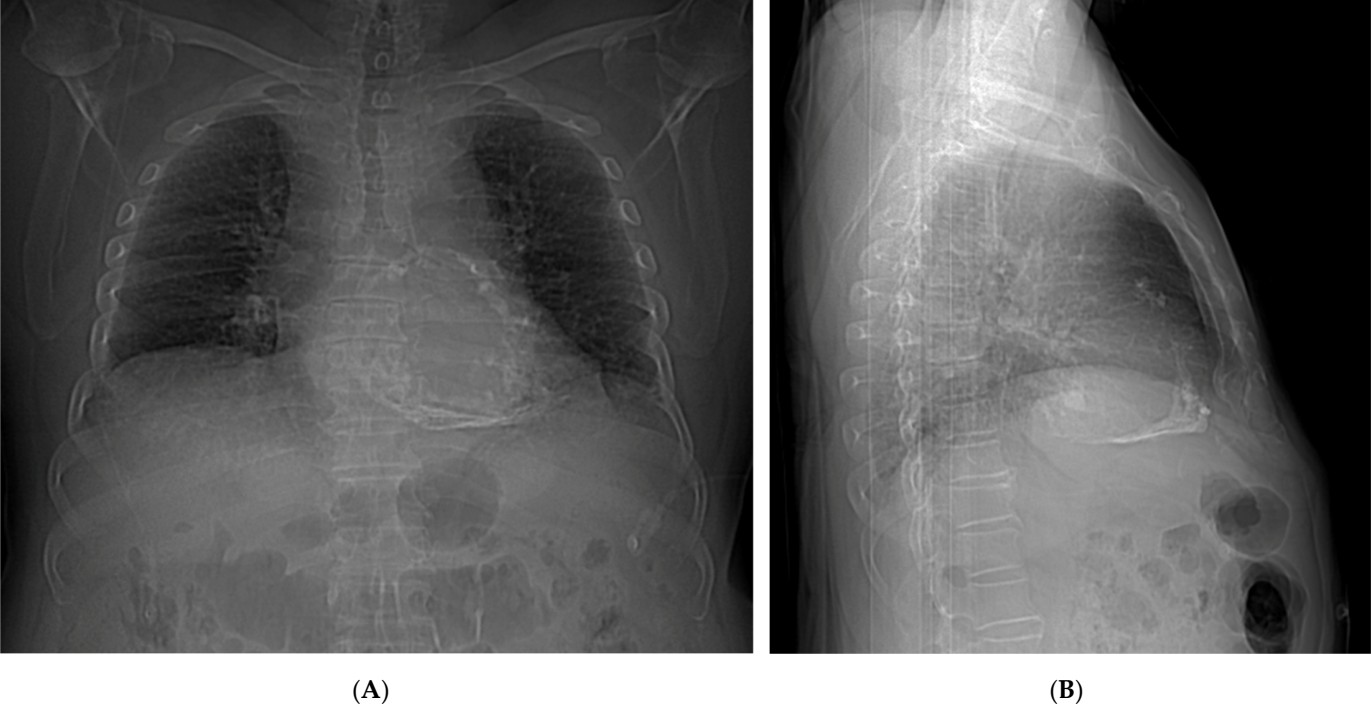

(**A**)  (**B**)

**Figure 1.** Chest X-ray and electrocardiogram. (**A**) Posterio-anterior chest X-ray showing pericardial calcification. (**B**) Lateral chest X-ray showing inferior wall pericardial calcification.

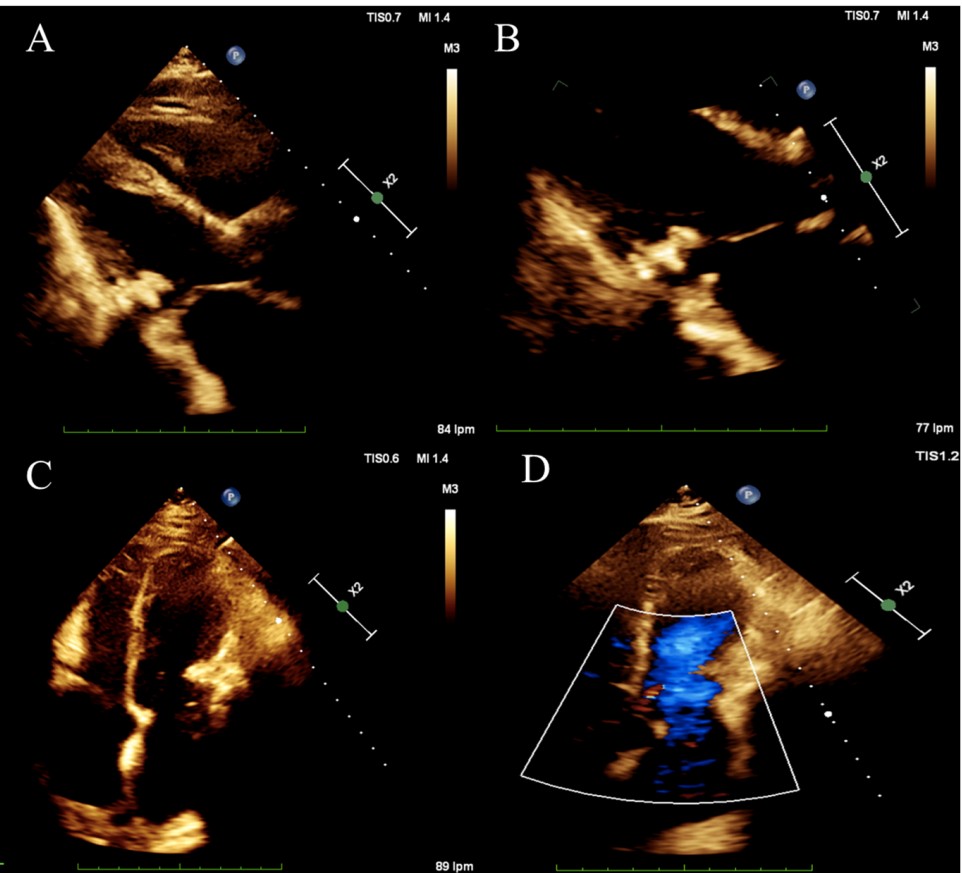

**Figure 2.** (**A**,**B**) Long-axis view showing pericardial calcification adjacent to mitral valve. (**C**) Apical four-chamber view with pericardial calcification in the anterolateral wall of the left ventricle. (**D**) Color Doppler four-chamber view showing no significant gradient in mitral valve.

Tuberculosis was ruled out. No evidence of acute or chronic infection was found, either from anamnesis or blood sample studies. Metabolic, autoimmune, and oncological causes were also investigated and further excluded as causative.

In this case, we consider the etiology of calcification to be idiopathic, although it could be related to ischemic coronary disease with post-infarction pericarditis, as the calcification was located mainly in the inferior, inferoseptal, and inferolateral segments, which are territories supplied by right coronary artery—where the patient previously had an infarct. Post-infarction pericarditis could explain how the calcification extended to adjacent territory perfused by circumflex coronary artery. Anterior and anteroseptal segments supplied by the left anterior descending coronary artery were mainly free of calcification.

The patient declined surgery after careful consideration aided by family and physicians. He was initiated on goal-directed medical therapy for heart failure with the significant improvement of exertional dyspnea. After two years of periodic follow-ups, there is no evidence of the progression of pericardial calcification or the worsening of signs and symptoms.

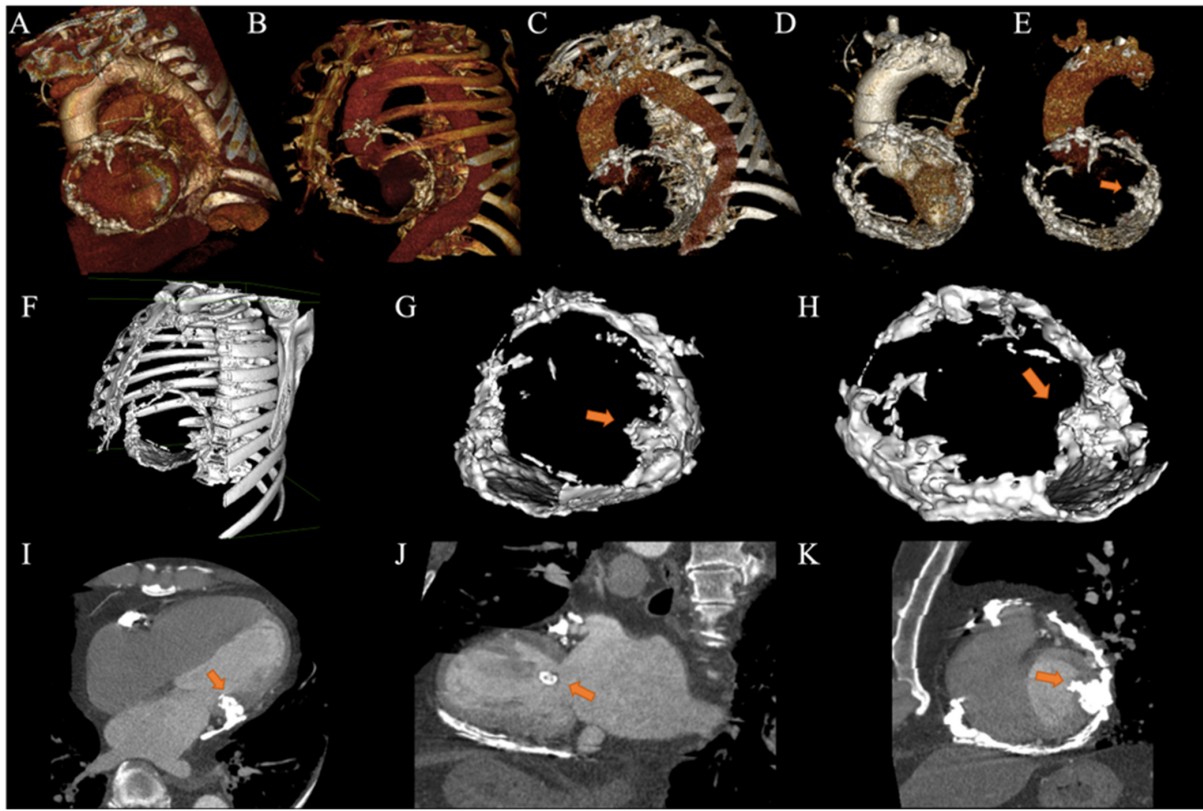

**Figure 3.** Cardiovascular CT. (**A–E**) Three-dimensional CT reconstruction showing pericardial calcification location and relation with the LV. (**F–H**) Three-dimensional CT surface rendering reconstruction showing calcification. (**I**) Four-chamber view showing pericardial calcification extending inside the left ventricular cavity. (**J**) Two-chamber view showing predominant inferior wall pericardial calcification. (**K**) Short-axis view.

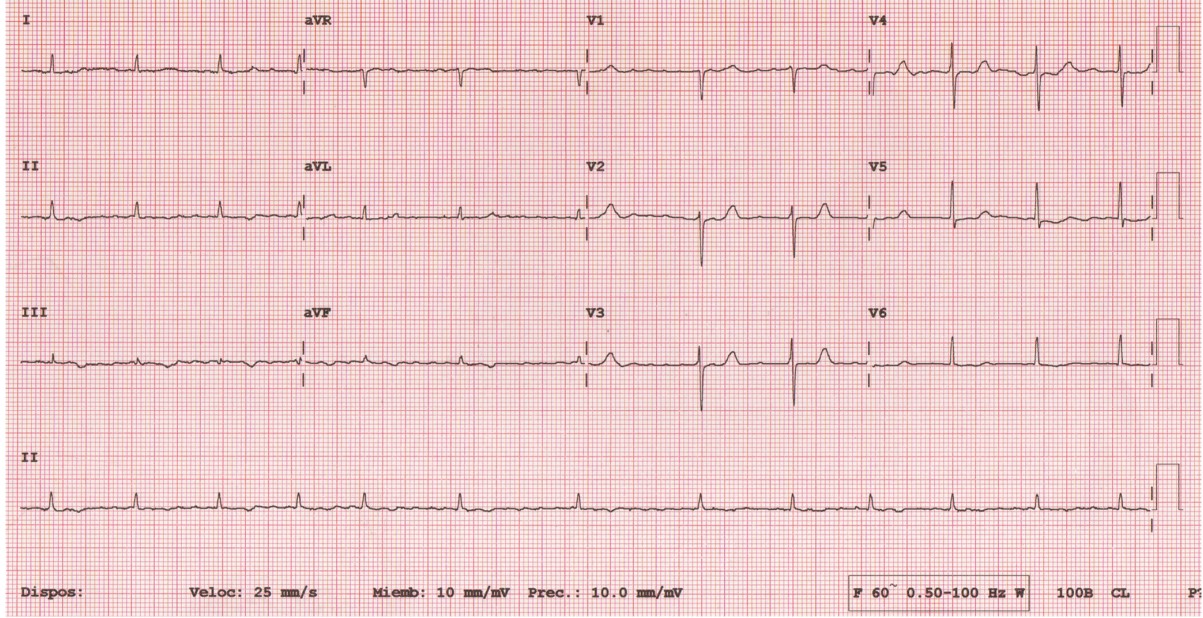

**Figure 4.** Electrocardiogram with Q-waves in leads I, aVL, V5, and V6; ST-segment elevation in leads I, V5, and V6; and ST-segment depression in leads III, V1, and V2.

## 3. Discussion

Pericardial calcification is a cause for heart failure, which can derive from inflammation and is mostly related to viral infections, radiation exposure to the chest, and post-heart surgery [1,2]. In the past, tuberculosis was considered a major cause of the disease; at present, it is still prevalent in underserved regions. However, it is now also known that asbestos exposure, injuries, cancer, rheumatoid arthritis, connective tissue disorders, and uremic pericarditis can facilitate calcification [3,4]. For the considered case, infectious processes were ruled out, along with oncological, rheumatic, environmental, and connective tissue disorders. Even when the trajectory of calcification concurs with a relatively recent stroke, it also may depict the route of catheterization four years after an ischemic event, indicating the cause as either post-infarction pericarditis, post-ischemic, or idiopathic. It is worth noting that around 50% of cases with pericardial calcification are considered idiopathic, according to the existing literature [6].

The diagnosis of constrictive pericarditis is based on characteristic hemodynamic and anatomical features and, in certain cases, can even persist undiagnosed. As seen in the present case report, clinical manifestations can be unspecific; for our patient, the chief complaint was dyspnea. Therefore, through the use of echocardiography, catheterization, cardiac MRI, and/or CT, pericardial calcification can be identified—either isolated or coexisting with other overlapping conditions [7]. Even though pericardiectomy resolves constrictive pericarditis and may lead to a full recovery [8], given the concomitant health conditions of our patient and their overall tolerance of the main symptoms, pericardiectomy was not considered as an ideal option. Pericardium resection due to calcification represents, according to a pathological series, around 36% of conditions related to pericardiectomy [9]. In our patient, resection could be reconsidered in the case of worsening dyspnea or syncope.

Atrial fibrillation has been reported in some cases of pericardial calcification to cause constriction; however, it has mainly been reported simultaneously with the finding of calcium deposits in the pericardium [10–13]. Notably, in the present case, atrial fibrillation was a pre-existing condition that may or may not be related to later calcium deposits.

## 4. Conclusions

Congestive pericarditis due to calcification can manifest as unspecific symptoms in patients with previous cardiovascular conditions, such as the one reported in this study. Combined imaging studies are crucial not only for identifying calcium deposits but also to assess a patient inherently prone to co-existing and exacerbating conditions. Even though pericardiectomy allows for the alleviation of the clinical manifestations of congestive pericarditis in the most symptomatic patients with pericardial calcification, in patients like ours, with tolerable symptoms, a cardiologist could discuss therapeutic options considering the patient's choices, potentially including a rehabilitation plan as part of non-pharmacological management.

**Author Contributions:** Conceptualization, M.S.-T.; methodology, N.M.-D. and K.S.-Z.; software, M.S.-T.; validation, E.S.-F. and N.M.-D.; formal analysis, M.S.-T.; investigation, N.M.-D. and A.G.-A.; resources, M.S.-T. and A.G.-A.; writing—original draft preparation, M.S.-T. and E.S.-F.; writing—review and editing, N.M.-D.; visualization, E.S.-F.; supervision, E.S.-F. and K.S.-Z.; project administration, M.S.-T.; funding acquisition, N.M.-D. All authors have read and agreed to the published version of the manuscript.

**Funding:** This research received no external funding, and the APC was funded by HRAEPY IMSS-BIENESTAR.

**Institutional Review Board Statement:** The study was conducted in accordance with the Declaration of Helsinki for studies involving humans and approved by the Ethics in Research Registry; number of the committee: CONBIOETICA-31-CEI-002-20170731 (protocol code: 2022-016 and approval date: 9 December 2022).

**Informed Consent Statement:** Written informed consent was obtained from the patient(s) to publish this paper.

**Data Availability Statement:** Supporting reported results can be found at ResearchGate from the authors' profiles.

**Acknowledgments:** We would like to acknowledge the medical and nursing staff who provided care for the patient from the present case report.

**Conflicts of Interest:** The authors declare no conflicts of interest.

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
