# Peer review of "Pericardial Calcification: An Uncommon Case with Intraventricular Extension"

_tomography, doi:10.3390/tomography10070076_

Round 1

Reviewer 1 Report

Comments and Suggestions for Authors

The authors present a case of significant pericardial calcification.  The images included are nice.

Comments:

1.  The "Gold Standard" for diagnosing pericardial constriction is full heart (left and right) catheterization.  Was this performed?  if so what is the result?

2.  The manuscript states "Diagnostic cathetherization was unremarkable."  Does that mean there is no pericardial constrction, or just no coronary artery disease?

3. What did past chest x rays and echocardiogram studies show?  ie. what is the progression of the calcification over time?

Comments on the Quality of English Language

Reasonable.

Author Response

COMMENT: Important: One reviewer missed catheterization; this seem to have been performed (bottom of page 5, but it is easy to overlook and I could not open it. This must be highlighted and made securely available. "The "Gold Standard" for diagnosing pericardial constriction is full heart (left and right) catheterization.  Was this performed?  if so what is the result?“

RESPONSE: Yes, it was, areas of infarction were found as in previous studies from the same patient, but no other changes were identified.

Images are not available, only scanning from screenshots unfortunately, we planned to include the report of the catheterization as a supplementary file, but we desisted since stent was functional and no new information was provided.

2.  The manuscript states "Diagnostic cathetherization was unremarkable."  Does that mean there is no pericardial constrction, or just no coronary artery disease?

RESPONSE. Dear reviewer, we apologize, we did not express well, we meant that no changes after the stent five years before

COMMENT: What did past chest x rays and echocardiogram studies show?  ie. what is the progression of the calcification over time?

RESPONSE: For the past 2 years chest x rays vary only slightly, but we have no access to all x rays together, unfortunately. Reports mention no significant change.

Reviewer 2 Report

Comments and Suggestions for Authors

Case report is interesting and well presented. The only comment that I would like to be clarified is on the sentence line 109 "the calcification was located along the right coronary artery territory". From figure 3 I and K the segment involved should correspond to segment 5/6 generally vascularized by the left circumflex artery according to Standardized Myocardial Segmentation and Nomenclature for Tomographic Imaging of the Heart (AHA Circulation 2002;105:539-542.). In this regard, how do the Authors hypothesize that the distribution territory of the right coronary is and that therefore the calcification can be attributed to the previous myocardial ischemia? Have they preliminarily studied the coronary circulation with conventional coronary angiography or with CT coronary angiography (supplementary material not found)? Please, clarify this.

Author Response

COMMENT: Case report is interesting and well presented. The only comment that I would like to be clarified is on the sentence line 109 "the calcification was located along the right coronary artery territory". From figure 3 I and K the segment involved should correspond to segment 5/6 generally vascularized by the left circumflex artery according to Standardized Myocardial Segmentation and Nomenclature for Tomographic Imaging of the Heart (AHA Circulation 2002;105:539-542.). In this regard, how do the Authors hypothesize that the distribution territory of the right coronary is and that therefore the calcification can be attributed to the previous myocardial ischemia? Have they preliminarily studied the coronary circulation with conventional coronary angiography or with CT coronary angiography (supplementary material not found)? Please, clarify this.

RESPONSE: Yes, Dear reviewer, what you mention is commonly correct, and the study you cite is greatly true, but according to reports, for the case of our patient, the territory partially was irrigated by the right coronary artery, by unremarkable we meant that no differences were reported compared to previous angiography five years before.

Authors want to apologize or not having all the images available, we included what was available, our hospital is a public third level health center providing care for the southeast region of the country. We performed and reported all studies you correctly mention, but unfortunately our country has undergone socio political transitions that have not exempted the Mexican health system; with these changes, public hospitals´ administrations have changed and with those changes, former contracts including those for digital archives changed from one moment to another, this has not only affected information for educational or scientific purposes, but also the monitorization of our known patients. Our patient lives 555 km from our hospital, we considered looking for personal they may have kept, but the patient and his family are unsure if they have kept copy of the studies.

Reviewer 3 Report

Comments and Suggestions for Authors

I have the following comments:

1) Is some more detailed information available wrt the STEMI occurred five years before? It would be especially interesting to know (e.g. from previous cardiac US or CT examinations) the status of the pericardium before or around the time of the STEMI, as this could help corroborating or ruling out an ischemic origin of pericardial calcifications.

2) From the CT images in Fig. 3, it seems that pericardial calcifications were distributed along multiple coronary territories outside that of the RCA. In particular, the bulky nodular calcification highlighted in Fig. 3K abuts transmurally the mid-lateral wall of the left ventricle (i.e., not the site of the previous inferior STEMI), which is usually perfused by diagonal and oblique marginal branches, i.e. LAD and Cx territories. What is your comments on that? It is actually odd that part of the otherwise supposedly healthy mid-lateral LV wall is so deeply involved due to an ischemic etiology.

3) Could it be hypothesized that pericardial calcifications were related to post-infarction pericarditis? Please provide some comments on this in the Discussion section.

4) Overall, the manuscript should undergo extensive English language editing, ideally by a native English-speaking medical writer or a professional editing service.

Comments on the Quality of English Language

Extensive English language editing required.

Author Response

COMMENT: 1) Is some more detailed information available wrt the STEMI occurred five years before? It would be especially interesting to know (e.g. from previous cardiac US or CT examinations) the status of the pericardium before or around the time of the STEMI, as this could help corroborating or ruling out an ischemic origin of pericardial calcifications.

RESPONSE: Five years before the mentioned consultation, patient had no visible calcification as it was not mentioned in any previous imaging studies reports

COMMENT: 2) From the CT images in Fig. 3, it seems that pericardial calcifications were distributed along multiple coronary territories outside that of the RCA. In particular, the bulky nodular calcification highlighted in Fig. 3K abuts transmurally the mid-lateral wall of the left ventricle (i.e., not the site of the previous inferior STEMI), which is usually perfused by diagonal and oblique marginal branches, i.e. LAD and Cx territories. What is your comments on that? It is actually odd that part of the otherwise supposedly healthy mid-lateral LV wall is so deeply involved due to an ischemic etiology. Could it be hypothesized that pericardial calcifications were related to post-infarction pericarditis? Please provide some comments on this in the Discussion section.

RESPONSE: Yes, dear reviewer, the classification you correctly mention is greatlyy true, but it is not accurate for all patients, also, other authors have provided diagrams with coexisting irrigations; RCA partially reached part of the LV wall and yes, we agree that pericardial calcifications could relate to post-infarction pericarditis, thus, we have included this in the discussion section. Thank you for this particular insight that we were not originally including.

Reviewer 4 Report

Comments and Suggestions for Authors

Thanks to the authors for their efforts in reporting this very interesting patient case. The report is clearly constructed from the patient's initial presentation until the end of the 2 years of follow-up. It illustrates an essential lesson: we are not dealing with images (particularly impressive in this patient) but with a set of clinical and investigative elements, without forgetting the elements provided by the discussion with the patient. This is clearly demonstrated in this observation, which deserves to be published.

But many elements are missing before publication: the initial clinical description is very poor: what were the clinical manifestations of this congestive heart failure : was there hepatomegaly, hepato-jugular reflux...What was the medical treatment (diuretic, ACE inhibitor, anticoagulation…)

The same for ultrasound. Your report has educational value and you must say what was expected from this Doppler ultrasound examination in decision-making, even if the patient was in atrial fibrillation thus preventing the collection of some signs of constrictive pericarditis described by Welch and al. that you cited (reference 5).

The same for hemodynamic

The same for metabolism, could you specify that serum calcium, serum phosphorus and protein levels were within normal values?

Finally, you introduce the term congestive insufficiency in the summary, introduction and conclusion. Moreover, you mention the link between constrictive pericarditis and congestive heart failure in the discussion in a very general way (lines 130-139) but without reference to the specific case of your patient.

Thank you therefore, after having better described the clinical and ultrasound hemodynamic signs of your patient, to state clearly in your patient the link between constrictive pericarditis (absent, moderate, moderate, severe) and congestive heart failure and the part of atrial fibrillation.

Author Response

COMMENT: the initial clinical description is very poor: what were the clinical manifestations of this congestive heart failure : was there hepatomegaly, hepato-jugular reflux...What was the medical treatment (diuretic, ACE inhibitor, anticoagulation…)

RESPONSE: Thank you, dear reviewer, we have corrected, please find corrected text highlighted

COMMENT: The same for ultrasound. Your report has educational value and you must say what was expected from this Doppler ultrasound examination in decision-making, even if the patient was in atrial fibrillation thus preventing the collection of some signs of constrictive pericarditis described by Welch and al. that you cited (reference 5). The same for hemodynamic. The same for metabolism, could you specify that serum calcium, serum phosphorus and protein levels were within normal values?
Finally, you introduce the term congestive insufficiency in the summary, introduction and conclusion. Moreover, you mention the link between constrictive pericarditis and congestive heart failure in the discussion in a very general way (lines 130-139) but without reference to the specific case of your patient.

RESPONSE: Thank you, dear reviewer, we have corrected, we added these aspects that were missing in our original version in the description and discussion, please find corrected text highlighted in the manuscript.

Round 2

Reviewer 1 Report

Comments and Suggestions for Authors

No further issues

Comments on the Quality of English Language

Satisfactory

Reviewer 3 Report

Comments and Suggestions for Authors

Thank you for your reply. No further comments.

Comments on the Quality of English Language

The English language of the manuscript should be mildly improved.

Reviewer 4 Report

Comments and Suggestions for Authors

Thank you, you answered all my questions and the article is now much better and can be published.